# Modeling Cyberattack Propagation and Impacts on Cyber-Physical System Safety: An Experiment

Théo Serru [1,2,*], Nga Nguyen [3], Michel Batteux [4] and Antoine Rauzy [5]

1  ETIS Laboratory-UMR8051, 95000 Cergy, France
2  Airbus Protect, 31069 Blagnac, France
3  Research Center, Léonard de Vinci Pôle Universitaire, 92916 Paris La Défense, France
4  IRT SystemX, 8 Avenue de la Vauve, 91120 Palaiseau, France
5  Department of Mechanical and Production Engineering, Norwegian University of Science and Technology, 7491 Trondheim, Norway
*  Correspondence: theo.serru@ensea.fr

**Abstract:** In this article, we present an experiment we conducted with discrete event simulations to analyze the effects of multi-step cyberattacks on the safety of cyber-physical systems. We show how to represent systems, their components (either software and/or hardware), communication links, security measures, and attacks from a malicious intruder. The latter are typically taken from the MITRE ATT&CK knowledge base. The discrete event simulation method makes it possible to represent any event affecting the system. We illustrate our approach by means of an illustrative example involving cyberattacks against the navigation system of an autonomous ship. We show how the formal modeling language AltaRica, primarily dedicated to safety analyses, can assess this illustrative example by representing the system and automatically extracting sequences of attacks, leading to a safety-critical situation, namely the deviation of the ship by the attacker. This article aims to discuss this approach and to outline the lessons learned from our experience.

**Keywords:** model-based security; cyber-physical systems; cyberattacks; safety; discrete event systems

## 1. Introduction

The boundary between the physical world and the cyber world is tenuous. Starting with computers, the cyber world has since expanded to smartphones, vehicles, the Internet of Things (IoT), cyber-physical systems (CPS), industrial control systems (ICS), etc. However, this relentless digitization also brings a rising danger, so-called cyberthreats. This is due to the expanding number of software and hardware components, which increase the spread of cyberattacks, while the growing connectivity between systems increases the attack surface, i.e., the number of entry points. Such cyberattacks may directly affect the safety of systems and, in the worse case, result in fatalities. To control the explosion of these risks and identify their impacts on security and safety, engineers usually perform security risk analyses (SRA). To help them in this job, formal languages and methods have been proposed to ensure that systems are safe and secure "by-design". In this assessment process, models usually take as input the architecture of the system and the results from the risk identification step. Then, by showing the impacts of cyberattacks, the architecture of the system evolves, and the analysis is performed once again. Thus, formal models applied to cybersecurity, or model-based security (MBS), aim to ensure that dependability and security attributes (i.e., safety, maintainability, availability, reliability, confidentiality, integrity, etc.) are protected in the case of cyberattacks. MBS needs to be scalable to real-world systems and comply with industrial regulations in order to ensure this. Surveys such as [1,2] show how current methods lack the tools and automation necessary to support automated security verification and validation. Indeed, as stated in [2], it is difficult, if

not impossible, to assess all the risks associated with cyberattacks by conducting security verification and validation manually.

The main contribution of this article is that it presents a model evaluating the impact of cyberattacks on the safety of CPS using a discrete event systems (DES) simulation. For this purpose, we detail the modeling process, define the formalism used, and provide a clear view of the outputs and the integration of the model in the design phase. More in detail, we demonstrate how to model systems using components (either software or hardware), communication links, security measures, and attacks from a hostile intruder. These attacks are based on the MITRE ATT&CK [3] knowledge base, and the DES enables us to model any event affecting the system. To give insight into the applicability of the approach, we develop an illustrative example of the navigation system of an autonomous ship inspired by several articles [4–7]. These articles allowed us to gather the inputs of the modeling process, which are the architecture of the system and the threats. We demonstrate how to model the navigation system and how to generate sequences of attacks. The autonomous vessel is then modeled using AltaRica, a formal modeling language based on DES and dedicated to safety evaluations. With this model, we use an existing tool to automatically generate all the sequences of attacks leading to a safety-critical situation: the deviation of a vessel from its route.

The remainder of this article is organized as follows. We first present the works related to our study in Section 2. Then, we detail the need for this work by depicting an illustrative example in Section 3. Section 4 establishes the formalism for cybersecurity analyses and outlines the main contributions of this work. Section 5 models an autonomous vessel with our definition of DES and conducts a qualitative analysis using the AltaRica formal modeling language and the OpenAltaRica tool. Finally, Section 6 concludes this work and discusses future perspectives.

## 2. Related Works

Model-based threat modeling for cyber-physical systems is the focus of this paper. Threat modeling (TM) has been developed in the last decades; it was initially developed for software development. This process focuses on the identification and analysis of threats related to the system under study [8]. On the other hand, model-based (MB) approaches use models as an artifact of the development process of a system [2] for threat analysis, illustration, etc. We believe MB and TM should come together to identify every attack threatening the system and to use automatic verification and validation from formal models. However, as outlined in [9], TM is a diverse field lacking a common ground, making the comparison between works more difficult.

This diversity comes from the construction of the approaches, which is organized around 4 parameters: the issue(s) the method aims to tackle, the threat model used (number of threats, level of abstraction, etc.), formalism, and the algorithm used for the validation/output generation. These aspects are the key to appreciate the diversity of the literature, as every work is built around a specific question using a specific threat model, formalism, and algorithm/tool. If these aspects are assessed in [9], it is done separately without linking them for each work.

Therefore, in this section, we propose to introduce some works that can answer our issue: generating sequences of attacks leading to a (safety) critical situation. We will introduce some common threat models and formalism and will not enter the algorithmic solutions, as they are not developed in this work (and as they depend on the three other parameters). To identify relevant works in this field, we can refer to the recent surveys [1,2,10] for CPS and [11] for SCADA systems. These surveys also identified challenges regarding threat modeling or models: the number of threats considered, the possibility to model physical and cyber-layers of CPS, the need for formal modeling, the modeling of non-functional requirements (such as safety), compliance with standards, the modeling of the architecture, etc. In addition, Xiong et al. [9] focused on TM and outlined the fact that approaches should

be more automated and that the authors should provide more assurance on the validation methods used.

### 2.1. Threat Models

First, we introduce some threat models commonly used in the literature. Among them, the STRIDE framework [12] has gained popularity and is now widely used for threat identification. This framework classifies threats in six categories: spoofing, tempering, repudiation, information disclosure, denial of service, and elevation of privilege; which gives a common classification for the TM process. It can be used without a model [4] or with dataflow diagrams (DFD) [13,14]. The last two articles use DFD to model the system with its components and dataflow and use STRIDE to identify potential threats that could affect them. STRIDE has also been used in [15] along with the Smart Grid Threat Modeling Tool (SG-TMT) to automate the process.

Other works use MITRE knowledge bases, e.g., ATT&CK [16] or CAPEC [17] as a threat model. For example, Xiong et al. [18] used MITRE ATT&CK as a basis for a domain-specific language (DSL) called enterpriseLang. This language is then used for attack simulation in two case studies. ATT&CK is also used in [19] to model attacks against ICS and generate attack scenarios using a hidden Markov model.

Ullah et al. [20] used ATT&CK along with the common vulnerability enumeration (CVE) and common weakness enumeration (CWE) to identify threats that could affect energy systems. CVE and CWE are low-level knowledge bases that try to cover all known vulnerabilities and weaknesses. In this direction, some works try to tackle the lack of common threat classification by mapping the different knowledge bases. An example is the work from A. Brazhuk [21], an ontology-based threat analysis integrating the well-known security enumerations (ATT&CK, CAPEC, CWE, and CVE). The major contribution of this work is to provide a database of threats that could be used in a model-based threat modeling process. Välja et al. [22] also developed an ontology-based framework to automatize the TM process and improve the quality of data used.

### 2.2. Formalisms

The threat models presented above are used to be implemented in models and benefit from their expressiveness to generate attack scenarios. Some of the first models that have been used in the literature are attack trees (AT) and attack graphs (AG), which were implemented back in the 1990s [23–25]. Nowadays, both of them are common in engineering, which eases their adoption. AT is a Boolean formalism allowing one to generate sequences of attacks from a tree model. Since this formalism is less complex than state-event formalism, it benefits from very efficient algorithms for cutoff generation. This efficiency comes at a cost as AT cannot consider reconfiguration, represent the architecture of the system or execute a single action multiple times. More importantly, resulting cutsets are sets of unsorted events. Because the order of attacks is crucial to understanding the attacker's behavior, this last element is the most problematic and has been tackled in the literature. The survey [26] illustrates the efforts to address the above-mentioned shortcomings for both AT and ADT (attack defense trees). Among the works, [27,28] integrated sequential conjunctions with the SAND gate and linear logic. In addition, [29,30] interpreted AT in timed automata to execute a single action multiple times and to use timed transitions and model checking tools.

AG is also a formalism dedicated to the discovery of attack paths. Some of the earliest works [31] employed model checking to build the attack paths but could not solve the state space explosion. Then, several works focused on lowering computation costs [32,33], on the automatic generation of AG [34], compact graphs [35], and probabilistic graphs [36], or on enhancing AG expressiveness [37,38]. Tools also resulted from the research effort, e.g., ADVISE [39], NuSMV [31], MulVal [40], etc. Today, after years of development, several challenges remain unsolved because of the limitations of AG. We can mention the scalability problems, the difficulties in modeling large graphs (and in reading them; Lallie et al. review

the different representations in [41]) and the inability to model system reconfigurations. Finally, as stated in [42], the main focus of AG assessments is on reachability analyses.

Other models have been used in the literature to analyze how the attacker can threaten a CPS. Among them, state-event formalisms are popular thanks to their abstraction capabilities and their use of time and probabilities. For example, we can see approaches using Markov chains [19,43,44] and stochastic Petri nets [45] or DES [46]. Each of these works are focusing on some challenges of model-based security, such as the formal modeling, the modelling of physical and cyber layers, or compliance with standards (with quantitative analyses).

Then, it is common to see algebraic representations in the literature to model concurrency between systems. The first works using algebraic representations [47,48] were focusing on analyzing the effects of cyberattacks on security attributes (confidentiality, availability, integrity, repudiation, etc.). More recent approaches, such as [49] or [50,51], translate AT into formal algebra and generate test cases or attack scenarios.

At last, surveys [1,2,10,11] have uncovered more works devoted to the generation of attack scenarios using ad hoc formalisms or languages that are not specifically designed for security analysis. For instance, Kang et al. [52] used the Alloy language to model a water treatment system and to identify cyberattacks that jeopardize the system's safety. This work was supported by a testbed to play the attacks identified and by Alloy Analyzer to generate attack scenarios. Alloy Analyzer is built on a "small-scope" analysis, which assumes that small counterexamples are sufficient to demonstrate software flaws. Such a concept might be effective while creating software, but it seems less pertinent when dealing with cyberattacks on CPS, where larger architectures lead to longer attack sequences. Li et al. [53] developed a security model based on SysML called SysML-Sec. It aims to model CPS and assess the communication protocol's security and relationship to safety. Then, since SysML is not a formal language, the tools ProVerif and UPPAAL are utilized to obtain the results. Using them allows verification on the model, but the translation between SysML and the tools' languages may be error prone. Finally, Zographopoulos et al. [54] decomposed the process into threat modeling, risk assessment, and system modeling. Through this decomposition, they were able to thoroughly examine CPS security and assess the real-time impact of cyberattacks on the system's operations. The article illustrates specific attacks and evaluates their impact on power distribution, with no mention of attack scenarios.

As stated in the Introduction, this article aims to ease the work of security engineers performing security risk assessments as part of the design phase. In this assessment, they try to identify every attack that can affect the system, their interrelations, and consequences on safety. Model-based assessment is an alternative to document-centric assessment, but the steps are the same. Given this, we want our model to be easily integrated into the security assessment process and to guide the evolution of the architecture towards a safer CPS. The architecture of the model and the list of threats affecting every component are the starting point of the modeling process. We shall use and represent them in a model to enable the generation of all the sequences of attacks (i.e., combinations of individual threats, or atomic attacks, towards a goal). Finally, we want this model to be formal and usable with existing language/tools to ease its adoption in industry. The presentation of related works in this section has shown the wide diversity of approaches leading to slightly different results. Only Kang et al. [52] had the same vision of the attack scenario, but their method might encounter limitations when generating longer sequences. In the remainder of this article, we will present a model focused on the engineers' needs and on the above-identified challenges, using discrete event simulation.

## 3. Illustrative Example: Autonomous Vessel

Instead of building an example for this paper, we decided to use an architecture and threats already studied in the literature. Therefore, the example we use is an autonomous vessel, inspired by [4–7]. In these articles, the authors warn that ships are at risk regarding

cyberattacks given that they are adopting more and more information and communication technologies (ICTs). Among these cyber-systems we can find navigation, positioning, identification systems, communications systems, integrated bridge systems control, electro-mechanical systems, and more [4]. This new generation of ships is called "cyber-enabled ships" (C-ES) and can refer to both remotely controlled ships and autonomous ships. In the articles [4–7], the authors identified the main threats in C-ES subsystems, which we will consider to generate attack scenarios (the following step in the assessment process).

### 3.1. Architecture

Katsikas et al. defined the architecture of an autonomous vessel in [4]. To build this architecture, they were inspired by MUNIN deliverables and the BIMCO report "The Guidelines on Cyber Security Onboard Ships" [55]. To maintain a relatively simple model with the possibility of generating numerous cyberattack scenarios, we focused on the navigation system of the vessel. According to Tusher et al. [56], the latter is the most vulnerable to potential cyber threats. The resulting architecture model is depicted as a graph in Figure 1 and is inspired by [5,7].

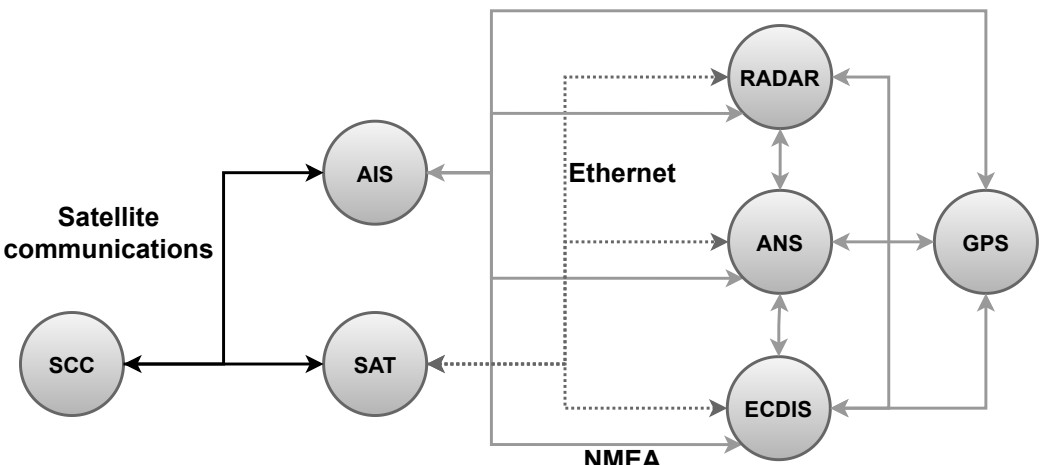

**Figure 1.** Architecture of the Autonomous Ship's Navigation System.

In Figure 1, the boxes represent sub-systems of the navigation system, and the links represent the communication links between them. We have one wireless link, the satellite communication, and two physical links, Ethernet and NMEA (named after the National Marine Electronics Association). The assailant can use these communication links to reach the system and its components. The considered nodes are:

- ANS: Autonomous Navigation System, which ensures the navigational functions of the vessel;
- GPS: Global Positioning System receiver, which receives the coordinates of the vessel;
- ECDIS: Electronic Chart Display and Information System, which transmits maps and other useful information;
- AIS: Autonomous Identification System, which provides information which, together with other systems, helps authorities and other ships to monitor sea traffic;
- RADAR: provides the bearing and distance of objects in the vessel's vicinity for collision avoidance and navigation at sea;
- SAT: satellite emitter/receiver, which allows a ship to communicate with other vessels and on-ground infrastructures;
- SCC: Shore Control Center, which communicates information about traffic with the ship.

In this architecture, the only components that can be attacked by a remote adversary (i.e., connected to the outside world) are AIS, GPS, RADAR and SAT. For AIS and SAT, the attacker can exploit vulnerabilities, enabling them to penetrate the system. Attacks against GPS or RADAR from the outside are commonly spoofing and jamming. They make

the system malfunction but are not entry points for penetration. Finally, to strengthen the system, some of these subsystems are embedding security measures, which are defined in the following subsection.

### 3.2. Security Measures

Security measures are essential in CPS. Without them, an attacker could perform any attack without difficulties. Thus, every (sub) system comes with a certain amount of security measures, which can be of different kinds (good practices, hardware reconfigurations, formal code verifications, intrusion detection systems, firewalls, etc.). In Table 1, we present the main security measures considered in this study. As there are a few of them, we explain in the next subsection about threats, which consider the system's architecture and security measures.

**Table 1.** Security measures of the autonomous ship's components.

| Component | Security Measures |
| --- | --- |
| AIS | Authentication, monitoring physical access |
| ECDIS | Authentication, monitoring physical access |
| ANS | Firewall, authentication, monitoring physical access, fail-safe procedures |
| Radar | Authentication, monitoring physical access |
| GPS | Monitoring physical access |
| SAT | Monitoring physical access |

### 3.3. Threats

Thanks to the definition of the architecture and security measures, we can now consider several threats the attacker may use to compromise the navigation system. They are taken from the original articles [4–7], from MITRE ATT&CK, or from other research works where the authors discovered new vulnerabilities [57–61].

Some of the considered threats are shown in Table 2 and are not all represented for a question of space.

**Table 2.** Example of Threats for the Autonomous Ship.

| Component | Threats |
| --- | --- |
| AIS | Eavesdropping, illusion, bogus information, sybil, impersonation, alteration/replay, masquerade, collusion, delay, timing attack [62,63], privilege escalation, identity spoofing, signal jamming [58] |
| ECDIS ANS | Exploitation of Apache vulnerabilities [59], malware installation during updates [60], denial of service (DoS), logic bombs, backdoors, SQL injection, data tempering, sensor freezing, obtaining control, erasing information [61] |
| RADAR | Denial of view [64], spoofing [60], dazzling |
| GPS | Jamming [65], spoofing [58], delay [66] |
| SAT | Penetration via satcom [58], malicious payload received via satellite |

### 3.4. Top Events

There are many threats that apply to an autonomous vessel, and most of them have a common goal, which is to deviate the ship from its original route. By doing so, the attacker can try to intercept, ground, or delay the ship in order to make profit, injure passengers, or alter the public image of the ship's owner. In the design process, safety analysis identifies top safety events, i.e., critical situations leading to harm toward the user or the environment, also called mishaps. For CPS deeply interacting with humans, it is critical to evaluate the effects not only of failures but also of cyberattacks on safety. Therefore, we can use the same top events resulting from analyses such as FHA (functional hazard analysis) or FMEA (failure mode and effect analysis) and focus on the risks for passengers by defining two

safety-critical events. First is the deviation of the vessel from its original route with a means to avoid collision (the radar). The second refers to the deviation from the original route without collision avoidance, which can lead the ship to going aground. The conditions leading to these situations are:

Top Event 1: ANS receiving erroneous data from ECDIS (erroneous map information), GPS (erroneous location), AIS (erroneous location of other ships) or radar (detection of "ghost ships").

Top Event 2: The conditions of TE1 coupled with the undetected loss of radar, preventing the ship from detecting obstacles.

Finally, to justify our choices regarding the conditions of the top events, we made some hypotheses:

**Hypothesis 1.** *The ANS can detect the loss of ECDIS, GPS and radar. It needs a continuous data flow from them and thus detects their loss.*

**Hypothesis 2.** *The navigation system comes with a "safe mode" that is activated if the ANS detects an unusual situation (such as that defined in H1). If the ship enters the "safe mode", an expert team will respond to the aggression with appropriate countermeasures, as proposed in [55].*

**Hypothesis 3.** *The attacker will penetrate the system either from AIS or from SAT and will not do both in a single attack. This hypothesis is justified (only) for this specific case because there is no interaction between the scenarios initiated from AIS and SAT. It will not make the result "worse" than a scenario with penetration from one subsystem or another. Thus, we consider that the scenarios avoided with this hypothesis are already covered by the other sequences, and thus, that there is no problem in dismissing them.*

*3.5. Scenarios*

With the above architecture and threats, we identify several scenarios prior to the modeling of the system. The interest is to generate them with the formalism defined in the next section to prove its relevance. Thus, some of the threat scenarios are:

From [60]: "*During an initial probe, the Naval Dome sent a virus-laden email over the ship's satellite link to the captain's computer, which is regularly connected to ECDIS for chart updates. During the very next chart update, the virus transferred itself to ECDIS, where it immediately installed itself and began to work. Once in place, the virus altered the vessel's position during a night voyage, deceiving the officer of the watch.*" From [60]: "*Naval Dome used the local Ethernet switch interface that connects the radar to ECDIS, the voyage data recorder, and the bridge alert system to successfully enter the radar workstation. After doing so, Naval Dome succeeded in deleting radar targets from the vessel's bridge radar screen, effectively blindfolding the vessel.*" From [67]: The attacker spoofs the GPS signal and fools the receiver to think it is somewhere else, leading to a deviation of the ship. The attacker can infect the SCC to access the ship via the AIS [6]. Then, it can send false information about ships in the vicinity to the ANS to deviate the ship from its position [4].

## 4. Formal Representation of Attack Propagation on CPS

In this section, we introduce the core of the article, an approach using DES to model the behavior of any system mixing physical/logical and hardware/software components. It also models their threats and analyzes the risks of losing one or several critical components and their impacts on safety.

*4.1. Definitions*

We shall represent CPS and cyberattacks on these systems by means of discrete event systems. A definition of DES is given in the following:

**Definition 1** (DES). *A DES is a five-tuple $\langle V, E, T, s_0, CS \rangle$, where:*

- *$V$ is a finite set of variables;*
- *$E$ is a set of events;*
- *$T$ is a set of transitions, i.e., of triples $\langle e, g, i \rangle$, where:*
  - *$e$ is an event from $E$;*
  - *$g$ is a Boolean condition on variables of $V$, called the guard of the transition;*
  - *$i$ is an instruction, i.e., a mechanism that modifies the current values of variables.*
- *$s_0$ is the initial state of the system, where a state is a valuation of all variables of $V$. Thus, $s_0$ is the initial valuation of all variables of $V$;*
- *$CS$ is a set of critical states $cs$ characterized by a Boolean condition on the values of the variables of $V$.*

In our representation of DES, variables take their values into a finite set of symbolic constants. They are used to represent:

- The access privilege of the attacker on a component, e.g., none, user or root;
- The capacity of the attacker to harm the confidentiality, integrity, and availability of an asset (a component or function that is critical from a functional or cybersecurity point of view);
- The security measures present on components and communication links and their status with respect to the ongoing attack, e.g., activated, deactivated, or bypassed;
- The presence of security breaches;
- More generally, any information of interest regarding an ongoing cyberattack.

In addition, an attacker will perform attacks (modeled as events) to change the state of the system and eventually reach a critical state. In practice, we consider only "atomic" attacks involving either only one component (local events such as a privilege escalation on a component) or one communication link (events that make it possible for the attacker to pass from a component to another, i.e., lateral movement) of the system.

Now, an attack execution is an alternated sequence of states and transitions $s_0 \xrightarrow{e_1} s_1 \cdots \xrightarrow{e_n} s_n$, where the $s_i$s are states of the system and the $e_i$s are names of events of $E$ involved in the transitions.

**Definition 2** (Formal description of an attack execution). *Let $M = \langle V, E, T, s_0, CS \rangle$ be a DES. Then the set of executions on $M$ is the smallest set such that:*

- *$s_0$ is an execution (where the attacker does nothing);*
- *If $\sigma : s_0 \xrightarrow{e_1} s_1 \cdots \xrightarrow{e_n} s_n$ is an execution, $n \geq 1$, and $\langle e, g, i \rangle$ is a transition that is enabled in the state $s_n$, then $\sigma \xrightarrow{e} i(s_n)$ is an execution.*

*An attack execution $s_0 \xrightarrow{e_1} s_1 \cdots \xrightarrow{e_n} s_n$ is successful if at least one of the states $s_i$s, $0 \leq i \leq n$, is critical, i.e., it satisfies $CS$.*

Note that, in practice, we are only interested in successful attacks $s_0 \xrightarrow{e_1} s_1 \cdots \xrightarrow{e_n} s_n$ that end in a critical state without going through any other critical state.

Finally, as our goal is to generate these executions automatically, an algorithm that performs a depth search on the possible events of the system until reaching a critical state can be developed. It takes the DES model as input and generates a list of successful attack sequences as output. It consists of a succession of steps:

1. Executing a transition with an unmarked event and its guard satisfied;
2. Changing the values of the variables involved in the transition's instruction;
3. Marking the event and storing it in an active sequence, then proceeding to step one unless:
   - A critical state is reached; then, it must store the sequence and return to the last iteration to execute another transition;

- No transitions are left; then, it must return go to the last iteration to execute another one.

This algorithm would allow us to store all the successions of events leading to a critical state. With DES, a transition can be executed as long as the guard is satisfied. Therefore, the sequences generated will cover all the combination of events leading to a top event.

The next subsection will present some specificities of the model, such as the possibility of modeling a cost for transitions and their levels of abstraction.

### 4.2. Approach Specificities

#### 4.2.1. Costs of Events

From an algorithmic perspective, searching for successful cyberattacks may suffer from the increase in the number of cyberattacks. A solution is thus to limit the exploration to cyberattacks involving a limited number of events or events performed with a limited amount of time and resources.

This can be achieved by associating an abstract cost with each event and defining the cost of a cyberattack as the sum of the costs of its events.

Note that the cost needs to be a scalar: it can be an element of any mathematical space $P$, e.g., a combination of time and resource measures, provided that $P$ is equipped with an internal associative binary operation $\oplus$ (to aggregate the costs along the sequences of events) and a partial order $\sqsubseteq$ over the space (to compare the cost of the current sequence with the chosen threshold). $\oplus$ and $\sqsubseteq$ must simply verify the monotony condition, i.e., $\forall x, y \in P$:

$$x \quad \sqsubseteq \quad x \oplus y$$
$$y \quad \sqsubseteq \quad x \oplus y$$

In this work, we do not discuss the relevance of using time and probabilities as a cost for events. Instead, we provide only qualitative results in the form of sequences of events to address risks related to cyberattacks on the system and identify their causes and consequences. This is equivalent to associating a cost of 1 for every event, and still allows us to limit the depth of the cyberattacks generated and the combinatorial explosion.

#### 4.2.2. Threat Model

In model-based engineering, the question of the threat model is paramount, as the exhaustiveness and expressiveness of the approach depend on it. With a low level of abstraction, the number of attacks to consider is very large and ends up in an overly complex model. On the other end, too much abstraction will come at the expense of the completeness and refinement of the model. THe literature does not provide a solution on which abstraction to choose. For example, some works use the common vulnerability and exposure (CVE) [68] repository and define attacks as vulnerability exploits. Others use a more abstracted view and model attacks as in STRIDE [69]. However, the levels of abstraction of both methods are very different. While CVE is low-level, with over 160.000 CVE recorded, STRIDE abstracts 6 types of attacks on computer systems. In this work, we choose a compromise between the two and model attacks as described in the MITRE ATT&CK [3] database.

Therefore, we use the database's (sub)techniques and model the behavior of the system when one or several of them are used. If a technique can be used by an attacker on a component of the system, we may model it. The pre-conditions shall be modeled as guards ($g$) and consequences as instructions ($i$), and the ATT&CK ID or name is the name of the event ($e$). The modeler can add any relevant variable in the guard or instruction or model events unrelated to ATT&CK but justified in the model (e.g., attribute update).

The next section will show how to model the vessel with DES and how we abstract cyberattacks as they are in ATT&CK.

## 5. Modeling the Illustrative Example

### 5.1. Modeling via DES

The autonomous vessel introduced in Section 3 has been modeled with the formalism described in the previous section. To model its architecture, we defined components and communication links. These components and links were abstract blocks embedding a certain amount of variables. Then, we initialized their variables and, finally, model events. In this section and the following, we model the system with information from the articles cited in Section 3 to build a likely architecture, identify cyberattacks, and generate scenarios. As one may see, the architecture differs from Figure 1. This architecture contains only the components and communication links where attacks have been identified (i.e., see Table 2). This choice was made on purpose to keep the example simple and easily explainable. Finally, as explained in Section 3.4, the goals of the attacker are to deviate the ship with or without the radar still being active. A representation of the example system is shown in Figure 2. In this figure, we represent the variables embedded in components and communication links with their initial values (i.e., at the state $s_0$ ) in a "box and arrows" manner.

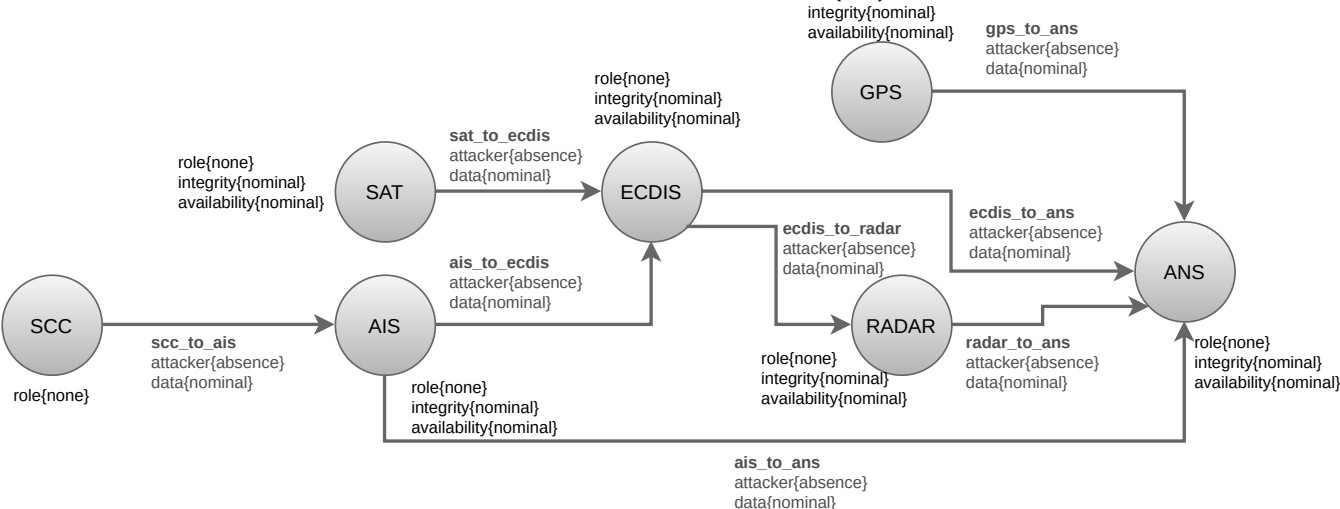

**Figure 2.** Illustration of the Autonomous Vessel Components and Communication Links with their Variables.

### 5.1.1. Modeling the System

In the model, we define the variables representing the attributes of the system's components and links. Thus, to understand which variable refers to which component or link, we use a specific syntax. For example, a variable referring to the *integrity* of the component *AIS* will be written *AIS.integrity*. This representation aims at making the model more readable and easier to build. Example 1 illustrates the approach by defining a simplified model with two components (SCC and AIS) and one communication link (scc_to_ais).

**Example 1** (AIS). *With our definition of DES, a model $M = \langle V, E, T, s_0, CS \rangle$ containing the SCC, the AIS and the link scc_to_ais would be formally defined as follows:*

- *The set V contains the variable referring to component SCC; this variable is {role} with*
  - *domain(role) = {none; user; root}.*
- *The variables of the component AIS are {role, integrity, and availability}, with:*
  - *domain(integrity) = domain(availability) = {nominal; loss; undetected_loss}.*

  *V also contains the variables of the communication link scc_to_ais, whose variables are {attacker and data}, with:*

- – *domain(attacker) = {presence; absence};*
  - – *domain(data) = {nominal, infected, erroneous, or none}.*
- • *The set of events E is composed of ATT&CK (sub)techniques: "attacker compromise scc" (T1584 Compromise Infrastructure), "attacker send infected update to ais" (T0843 Program Download), "malware deploys on ais" and "privilege escalation on AIS" (T1068 Exploitation for Privilege Escalation).*
- • *The set of transitions T is then:*
  - – $t_1$
    *e: attacker_compromise_scc*
    *g: (scc.role = none)*
    *i: scc.role ← root*
  - – $t_2$
    *e: attacker_send_infected_packet_to_ais*
    *g: (scc.role = root)and(scc_to_ais.data = nominal)*
    *i: scc_to_ais.data ← infected*
  - – $t_3$
    *e: malware_deploys_on_ais*
    *g: (scc_to_ais.data = infected) and (ais.role = none)*
    *i: ais.role ← user*
  - – $t_4$
    *e: privilege_escalation_on_AIS*
    *g: (ais.role = user)*
    *i: ais.role ← root.*
- • *The initial state $s_0$ is the following:*
  *scc.role = none; ais.role = none; ais.integrity = nominal; ais.availability = nominal; scc_to_ais.attacker = absence; scc_to_ais.data = nominal.*
- • *Finally, the critical state cs could be:*
  - – *ais.role = root.*

Now, let us explain some events and their associated transitions (their guard and consequences) in a non-formal way. **attacker_send_infected_packet_to_ais**: In this attack, the attacker uses its privilege on the SCC to send malware on the AIS. Thus, the communication channel does not send nominal data anymore but instead sends an infected packet.

**malware_deploys_on_ais**: To deploy a virus on the AIS, the communication link scc_to_ais has to carry a packet infected with a virus, and the role of the attacker is none on the AIS, meaning that it did not reach this component yet. As a result, the attacker accesses the component and obtains user privileges.

**privilege_escalation_on_AIS**: This event is only modeled if a vulnerability exists on the AIS. Then, we consider that the attacker will exploit it and needs access to the component (role = user). The consequence is the gain of root privileges.

5.1.2. Modeling the Top Events

The goal of our analysis is to evaluate the impact of cyberattacks on the safety of CPS. Top safety events are critical situations impacting the humans or the environment of the system. In our model, we define them as a Boolean condition on the variables of V. To do so, we must map the top events with the variables that define the state of the system: data, integrity, or availability. Then, expertise is needed for two steps: defining the system's state regarding components' state and defining the components' state with the security attributes.

In this article's example, we consider that the vessel will deviate from its original route if the ANS is fed with erroneous information, making it misbehave. Thus, the corresponding top event could be met if the incoming data are erroneous: $(ecdis\_to\_ans.data = erroneous)or(ans\_to\_ans.data = erroneous)or(radar\_to\_ans.data = false\_obstacle).$

### 5.1.3. Modeling Attacks

For this example, we model the fact that the attacker enters the system via the AIS or via SAT by modeling events that are enabled at the initial state (i.e., their guard is satisfied at $s_0$ ). Here, the initial events from the attacker are sending an infected message targeting the ECDIS via satellite [60] or sending an infected message to the AIS via the SCC [6]. The attacker can also spoof or jam the GPS signal, but this will not grant access to the system.

Then, several attacks are modeled for each component and links. These attacks are part of a scenario already identified (e.g., [60]) or are atomic attacks identified in Table 2. They can be categorized regarding their effects on the variables: privilege escalation (role) or (undetected) loss of integrity or availability for components and lateral movement for communication links.

In Section 4.2.2, we highlight the use of the ATT&CK knowledge base. Here, we abstract vulnerability exploits by their ATT&CK category, e.g., the exploitation of an Apache vulnerability to elevate privilege [59] is abstracted by ATT&CK T1068: exploitation for privilege escalation.

### 5.2. Remark on Confidentiality and Scalability

With the formalism we have described, one can model any variable of interest and model complex systems. In our case, we did not model a variable, "confidentiality", as we considered that the confidentiality of the information shared between the components of a CPS is not safety-critical. However, the model is highly flexible, as we could add a variable for confidentiality to model the impact of cyberattacks on the confidentiality of data.

To model more complex systems, more components and events can always be added. The only limit is the time that a modeler can spend on the model and the time needed to generate the results. Here, we illustrate the approach and show how to model a system and its cyberattacks, but we discuss scalability further in Section 5.3.

Generation of Attack Sequences

The goal of this experiment was to show how to generate the sequences of events the attacker can perform to affect the integrity of ANS. An example of such sequences is shown in Figure 3.

To automatically generate these sequences, we had two choices: develop a generator or use an existing program/tool. Such automated tools or formal languages already exist to model DES and generate sequences of events leading to a top event.

In this work, we chose to explore these tools and evaluated their relevance regarding our requirements. If none of these tools allowed us to model our example or to generate attack scenarios, we would have developed our own. However, we identified a language that allowed us to model the autonomous vessel and a tool to generate attack scenarios. This language is called AltaRica, and we explain, in the next section, the reasons for this choice.

| Top Event 1 | Top Event 2 |
|---|---|
| Scenario 1 | Scenario 1 |
| GPS.gps_spoofing | SAT.receives_email_with_infected_payload |
| gps_to_ans.gps_sends_false_position_to_ans | sat_to_ecdis.infected_payload_transfered_to_ecdis |
| ANS.ship_deviates_from_route | ECDIS.virus_alters_maps_during_chart_update |
| Scenario 2 | ecdis_to_ans.tempered_maps_information_sent_to_ans |
| SAT.receives_email_with_infected_payload | ANS.ship_deviates_from_route |
| sat_to_ecdis.infected_payload_transfered_to_ecdis | ecdis_to_radar.lateral_movement_from_ecdis_to_radar_via_ethernet |
| ECDIS.virus_alters_maps_during_chart_update | RADAR.attacker_accesses_radar |
| ecdis_to_ans.tempered_maps_information_sent_to_ans | RADAR.attacker_deletes_radar_targets |
| ANS.ship_deviates_from_route | radar_to_ans.deactivated_radar_send_working_signal |
| Scenario 3 | Scenario 2 |
| scc_to_ais.corrupted_scc_sends_malicious_package_to_ais | GPS.gps_spoofing |
| AIS.malware_deploys_on_ais | gps_to_ans.gps_sends_false_position_to_ans |
| AIS.exploitation_for_privilege_escalation_T1068_on_AIS | ANS.ship_deviates_from_route |
| ais_to_ans.attacker_sends_false_information_to_ans | scc_to_ais.corrupted_scc_sends_malicious_package_to_ais |
| ANS.ship_deviates_from_route | AIS.malware_deploys_on_ais |
|  | AIS.exploitation_for_privilege_escalation_T1068_on_AIS |
|  | ais_to_ecdis.lateral_movement_from_ais_to_ecdis |
|  | ECDIS.attacker_accesses_ecdis |
|  | ecdis_to_radar.lateral_movement_from_ecdis_to_radar_via_ethernet |
|  | RADAR.attacker_accesses_radar |
|  | RADAR.attacker_deletes_radar_targets |
|  | radar_to_ans.deactivated_radar_send_working_signa |

**Figure 3.** Example of Scenarios for Top Events 1 and 2.

*5.3. Implementation in AltaRica*

The previous section highlights the need for a tool to create and analyze DES and to automatize attack scenario generation. As our experiment is based on DES, we searched for existing tools and (formal) languages to model and analyze them. Furthermore, we are interested in the impact of cyberattacks on the safety of CPS; thus, we first searched for formal languages used in safety analyses and based on DES. Therefore, in this paper, we chose the language AltaRica [70] to model the system and its security properties.

The reasons for this choice are threefold. First, this formal modeling language is built on DES and is mostly used to model systems along with their safety properties. Thus, as the approach defined in this work uses a standard definition of DES, such a definition is already handled by AltaRica.

Second, some authors are working with AltaRica and are developing tools to model complex and interrelated systems along with their safety properties and to analyze them. Therefore, we chose this language and modeled the autonomous vessel in the third version, AltaRica 3.0 [70]. This enabled us to use the tools implemented in the OpenAltaRica platform to automatically generate the sequences of attacks and to evaluate the validity of the model.

Finally, AltaRica has been used in industry to model large systems with numerous components. For this reason, we assume that the formalism will be able to handle complex CPS.

As we just explained, AltaRica is a modeling language used in safety, and the tools developed are made for safety analyses. However, as CPS have potential effects on the loss of human lives, their cybersecurity risk analyses need to perform qualitative and quantitative assessments evaluating safety impacts. Then, if the security mechanisms are correctly modeled, OpenAltaRica can generate the sequences of attacks leading to a top safety event. The generator runs through the executions according to a target property (here, referring to one top event or the other) and the constraints of the model. Then, it generates all the executions leading to the target. One particularity of DES is the fact that the attacks can come from any component infected by the attacker. It results in a large number of sequences (i.e., attack scenarios), with the atomic attacks (use of a single threat) being launched in any order. To model the system in AltaRica, the variables of the model (introduced in Section 4.1) are defined as variables in AltaRica; events and transitions already exist in the language and are used. One major difference between AltaRica and the already defined formalism is that the first comes with the possibility of expressing timed transitions, which differs from the notion of the cost of events (defined in Section 4.2.1). However, we do not use timed transitions, as we consider that the approach defined in Section 4.1 suffices to obtain cyberattack scenarios regardless of time. Anyone willing to add timing to the transitions could do so using AltaRica.

### 5.3.1. Modeling of the Vessel

Here, we explain how we model the ship with AltaRica 3.0. A fragment of the model is given in Listing 1. In this example, we show the component AIS and the communication link from AIS to ECDIS, where a lateral movement can be initiated by the attacker.

In the components (e.g., AIS in Listing 1), the *role* is modeled as a *state* variable, a variable updated in transitions and initialized at $s_0$. A transition in AltaRica is also built with a guard and an instruction. State variables are thus used to express the inherent attributes of *components* and *communication links*, e.g., *role*.

To propagate information from one component to another, AltaRica uses a second type of variable, the so-called *flow variable*, updated via an assertion. Flow variables allow to communicate between components and then to propagate the value of variables (e.g., from AIS to ais_to_ecdis).

To consider communication links between components, we modelled them as *classes* that take as input the role of the attacker on the source component, allow to execute a transition of type *lateral movement*, and output the state of the data and the presence of the attacker.

For example, the value of the flow variable *vf_role_in* of the component *ais_to_ecdis* is used in the transition's guard associated with the event *lateral_movement_from_ais_to_ecdis*. The transition's instruction updates the state variable *vs_attacker* (*attacker*) of the same component. Finally, two flow variables *vf_data_out* and *vf_attacker_out* take the values of the state variables, and their values are propagated to the ECDIS.

### 5.3.2. Generation of Sequences

A cyberattack generated from an AltaRica model is a combination of states *s* and events *e*, $(s_0 \xrightarrow{e_1} s_1 \cdots \xrightarrow{e_n} s_n)$, as defined in Section 4.1. To generate sequences, the tool computes the sequences and prints out the ones validating a certain condition. This condition is given by a variable called *Observer*, which is true when one of the top events is reached.

In this work, we are interested in the attacks needed to compromise the system's safety. Thus, we do not want to generate sequences with attacks that do not help to reach the top event or scenarios with the same attacks, but in a different order (where the order has no impact). The sequences that satisfy these criteria are called **minimal sequences** and are the sequences we are looking for. However, the OpenAltaRica sequence generator does not generate minimal sequences. It rather generates all sequences leading to one of the unwanted states according to the already mentioned properties of DES. It thus results in many sequences, 40 for TE1 and 593 for TE2, generated in a second, with two to ten events.

The automation of the generation is a significant step, as analysts could not generate such numerous and long sequences manually.

**Listing 1.** Fragment of the Case Study in AltaRica 3.0: AIS and link ais_to_ecdis.

```
1  domain Role{none, user, root}
2  domain Data{nominal, none, infected, erroneous, false_nominal}
3  domain CIA{nominal, loss, undetected_loss}
4  class AIS
5  //Inputs
6    //SCC
7        Boolean vf_attacker_scc_in(reset=false);
8        Data vf_data_scc_in(reset=nominal);
9  //Internals
10       Role vs_role(init=none);
11       CIA vs_integrity(init=nominal);
12       CIA vs_availability (init=nominal);
13       CIA vs_confidentiality(init=nominal);
14 //Outputs
15       Role vf_role_out (reset=none);
16       CIA vf_integrity_out(reset=nominal);
17       CIA vf_availability_out(reset=nominal);
18       CIA vf_confidentiality_out(reset=nominal);
19 //Events
20       event malware_deployed;
21       event exploitation_for_privilege_escalation_T1068_on_ais;
22 //Actions
23  transition
24       malware_deployed: (vs_role == none) and (vf_attacker_scc_in == true) and (
         vf_data_scc_in==infected) -> vs_role := user;
25       exploitation_for_privilege_escalation_T1068_on_ais: vs_role == user -> vs_role := root;
26  //Assertion
27  assertion
28       vf_role_out := vs_role;
29       vf_integrity_out := vs_integrity;
30       vf_availability_out:=vs_availability;
31       vf_confidentiality_out :=vs_confidentiality;
32 end
33
34 class ais_to_ecdis
35 //Input
36       Role vf_role_in (reset=none);
37       CIA vf_confidentiality_in(reset=nominal);
38       CIA vf_integrity_in(reset=nominal);
39       CIA vf_availability_in(reset=nominal);
40 //Internals
41       Data vs_data (init=nominal);
42       Boolean vs_attacker (init=false);
43 //Outputs
44       Boolean vf_attacker_out (reset=false);
45       Data vf_data_out (reset=nominal);
46 //Event
47       event lateral_movement_from_ais_to_ecdis;
48 //Actions
49  transition
50       lateral_movement_from_ais_to_ecdis: (vs_attacker == false) and (vf_role_in==root) ->
         vs_attacker := true;
51  assertion
52       vf_attacker_out := vs_attacker;
53       vf_data_out := vs_data;
54 end
```

In comparison, in the model of Section 5.1, we identified 6 minimal sequences for TE1 and 25 for TE2 (involving the scenarios identified in Section 3). This shows the need of minimality to improve the generation.

Among minimal sequences, the smaller ones for TE1 contained only one action from the attacker, i.e., GPS spoofing. The smaller ones for TE2 contained four attacks, i.e., an infected payload sent by email via satellite, a virus altering maps during an update, lateral movement from ECDIS to RADAR, and radar target deletion. All sequences generated were successful and lead to TE1 or TE2. The sequences generated with OpenAltaRica were the same as the ones shown in Figure 3.

One may see that with a relatively small example, the number of sequences generated can be relatively large. In the meantime, the computation time is very low, but it will increase with a larger and more complex system. This complexity is due to the possibility

of launching an attack from any component already infected, made possible by the use of DES. Some ideas to tackle this issue are discussed in Section 6, as this is decisive in the scalability of this approach.

To conclude this section, we showed that AltaRica allows us to model an autonomous vessel regarding the model of Section 5.1. The choice of the AltaRica formal modeling language was motivated by the possibility of automatically generating the scenarios containing the names of the consecutive events. Automatic generation is of critical importance to enable this approach to be used in industry. It gives the information we need to identify attack paths and secure the system. Some improvements are needed to make the result exploitable and will be highlighted in the last section.

## 6. Conclusions

In this study, we conducted an experiment to support the security risk assessment of cyber-physical systems. This approach is based on discrete event systems and analyzes the effects of cyberattacks on the safety of CPS. It does so by describing systems via their hardware and software components, communication links, security measures and events (cyberattacks or system reconfigurations), which are based on the MITRE ATT&CK knowledge base. A model can be used to generate sequences of events that result in a safety-critical situation by expressing how the system would behave in the case of a cyberattack.

Then, we introduce the AltaRica formal modeling language, which enables us to use analysis tools to run automatic qualitative and quantitative assessments from a model. An autonomous ship illustrative example has been modeled with DES and AltaRica to illustrate the benefits of the approach. Then, the automatic generation of attack scenarios has been performed using the OpenAltaRica platform.

The article demonstrates how modeling a CPS architecture with DES and AltaRica is achievable, as well as how to exhaustively identify the ways an attacker could threaten the system. We are optimistic that by reducing human error and enhancing maintainability, our approach can be useful for engineers working on security risk assessments. However, we still have to answer several issues, which leads us to future perspectives.

The approach was built upon DES to describe systems and ATT&CK to define attacks. Such a choice led to a "medium" level of abstraction and many sequences generated. As illustrated in Section 5.3, the number of sequences of events generated from the model is large, and it might take time to analyze them one by one manually. Indeed, with DES, as long as a transition is executable, it may be executed. Consequently, we may define what is an attack scenario in terms of its constitutive elements, considering that the attacker will not perform unnecessary actions. Such a definition would enable us to reduce the generated sequences and ease the post-processing filtering. A solution might be to consider the causality relations between attacks (modeled as events) instead of the number of attacks in the scenario (as introduced in Section 4.2.1). This last filtering method works well in safety with independent events but might be less relevant when studying cyberattacks. Then, we could consider weighting sequences of unrelated attacks, whether they are long, and not weight sequences of attacks executed in a logical order. This solution has been explored and illustrated with an industrial case study in the authors' article [71].

Secondly, the modeling of only a few numbers of threats can lead to a non-exhaustive analysis and, thus, sleeping vulnerabilities inside the system. This implies the use of another tool or method to cover a maximum number of threats. Even in this case, it is not possible to ensure that the system is protected from 100% of the threats. In addition, regulations do not help to answer this lack of generic view, as they are mostly process-oriented and domain-dependent. Thus, in this work, we chose a level of abstraction coherent with that of ATT&CK techniques and allowed the opportunity for the applicant to model every attack. This choice has thus to be validated regarding large real-world case studies to ensure that we can consider more complex and interrelated attacks. Additionally, the creation of an open-access knowledge base for CPS-related cyberattacks, such as the recent addition in MITRE ATT&CK regarding ICS, might be a good start to generate a common threat model.

Finally, it may be interesting to compare the related works on a common case study. Such a comparison will allow us to evaluate the capabilities of the proposed formalisms and see what is the best compromise between expression power and algorithmic complexity.

**Author Contributions:** Conceptualization—methodology, T.S., N.N., M.B. and A.R.; supervision N.N., M.B. and A.R.; writing, T.S. All authors have read and agreed to the published version of the manuscript.

**Funding:** This research was funded by the CY Initiative Excellence and Airbus Protect.

**Institutional Review Board Statement:** Not applicable.

**Data Availability Statement:** Not applicable.

**Acknowledgments:** We would like to thank Laurent Sagaspe, Raphael Blaize and Emmanuel Arbaretier for their support of this work.

**Conflicts of Interest:** The authors declare no conflict of interest.

## Abbreviations

The following abbreviations are used in this manuscript:

| | |
|---|---|
| IoT | Internet of Things |
| CPS | Cyber-Physical Systems |
| ICS | Industrial Control System |
| SRA | Security Risk Analysis |
| MBS | Model-Based Security |
| DES | Discrete Event Systems |
| ICT | Information and Communication Technologies |
| ANS | Autonomous Navigation System |
| GPS | Global Positioning System |
| ECDIS | Electronic Chart Display and Information System |
| AIS | Autonomous Identification System |
| SAT | Satellite |
| SCC | Shore Control Center |
| TE | Top Event |
| CVE | Common Vulnerability and Exposure |
| TM | Threat Modeling |
| MB | Model-Based |
| DFD | Dataflow Diagram |
| SG-TMT | Smart Grid Threat Modeling Tool |
| DLS | Domain-specific Language |
| CWE | Common Weakness Enumeration |
| AT | Attack Tree |
| AG | Attack Graph |
| SysML | System Modeling Language |
| C-ES | Cyber-enabled Ships |
| NMEA | National Marine Electronics Association |
| FHA | Functional Hazard Analysis |
| FMEA | Failure Mode and Effect Analysis |

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
