# Peer review of "Modeling Cyberattack Propagation and Impacts on Cyber-Physical System Safety: An Experiment"

_electronics, doi:10.3390/electronics12010077_

Round 1

Reviewer 1 Report (New Reviewer)

The paper provides an example of the application of modelling techniques to describe cyberattacks concerning a ship using Discrete Event System modelling. The authors have adopted significant reductions in the complexity of a real-world application, by considering just a subset of the possible events, a subset of the possible threats, and assuming that simplifying models of the system. Its value lies in being illustrative of how a modelling effort can be undertaken, but does not provide general indications. The title is quite clear on that, but the authors should stress it in the Introduction. Of course, this strongly limits the value of the paper.

Please find hereafter my more detailed comments:

1) the paper heavily relies on Refs. [4-7] to build the case model. The authors should clearly state what is new and original with respect to those papers, or if it is just an illustration of models and techniques described in those papers.

2) The initial subsection of Section 3 (Section 3.1 "Introduction") could dispense with the title. Its text could just run as an introductory text right after Section 3's title

3) In Section 3.2, in addition to the description of the nodes in the graph of figure 1, the authors should also add a description of the relationships described by the edges of the graph.

4) In Section 3.3, the authors should include a brief description of the security countermeasures present in each node of the graph, rather than just providing an example for a single node.

5) The authors do not provide any indication on how to select top events in Section 3.5. If the system is complex, identifying the full set of top events may be difficult. How do you carry it out? By elicitation of experts? By exploration of all the possible interactions through the graph? Though this is just an example, some indication of how a general procedure may be applied would be needed.

6) In the specific case, hypothesis 2 in Section 3.5 may be a bit lacking. It assumes that entering the safe mode would bring the ship to a stop with no further damage. Actually, this is hardly true in real ships, since the stopping time of ships is by no means close to zero (it can also reach half an hour). During that time, the ship would continue its navigation and could anyway be subject to collisions. And let alone the case for military ships, where stopping would not guarantee the absence of further damages.

7) In Section 6.1, it looks like the only transitions considered are those where the initial state is a "guard" state. Also, the left-hand side of the transition is defined as a Boolean condition, while it should be a state (conditions trigger transitions). The authors should provide a clearer and more comprehensive definition of possible transitions.

8) In Section 4.2, it appears that the cost of events is anyway 1 (line 361). If that's the case, it is a very rough model of reality. Events are by no means equally dangerous, and their cost may be quite different. The authors should clearly state if that is a limitation of the approach or a simplifying choice of their own. Also, some examples of cost aggregation operations should be given.

9) The remarks on confidentiality and scalability scattered around Section 5.1 should be grouped together and reported as limitations of the approach.

10) In Section 5.2, the authors should provide some indication about the scalability of the modelling properties of Altarica.

11) Since the authors employ first an abstract formalism (sort of pseudocode) and then employ a specific modelling language, they should provide some indication about possible discrepancies, i.e., some operations that may not be possible under a specific modelling language.

12) Structuring the Conclusions into subsections is not needed.

Author Response

Dear reviewer,

Please find attached our answers to the comments.

Reviewer 2 Report (New Reviewer)

In this paper, the authors present an experiment they conducted with discrete event simulation to analyze the effects of multi-step cyberattacks on the safety of cyber-physical systems. In the first two sections, the authors describe their research background and related works. In the third section, they present an illustrative example: the autonomous ship. The presentation contains system architecture, security measures, threats and top events. In the fourth section, the authors show the formal representation of attack propagation on cyber-physical systems. They model this example via discrete event simulation in the fifth section. The last section contains their conclusions and plans.

This article considers a fundamental issue, which is the security of cyber-physical systems. Each component of cyber-physical systems should be verified in case to ensure the appropriate security level by them. Also, cyber-physical systems have different architectures, so they should be considered in many aspects because both security and vulnerabilities still evolve. The authors show how to exhaustively identify how an attacker could threaten the system, which is the most significant advantage of this article.
I have only one remark. The authors should exhaustively summarize and highlight their scientific contributions in the Introduction.
I suggest this article for publication.

Author Response

Dear reviewer,

Please find attached the answers to the comments.

This manuscript is a resubmission of an earlier submission. The following is a list of the peer review reports and author responses from that submission.

Round 1

Reviewer 1 Report

This is an interesting paper that adds a novel contribution to the field of cybersecurity. The material provides a sound approach for security modeling with discrete event simulation. They analyze the effects of cyberattacks on the cyber-physical system security using the formal modeling language AltaRica, represent the sample of the ICT system of an unmanned vessel, and extract conclusions on attack analysis. While the contribution of the research work is acknowledged, there are few drawbacks that make this contribution weaker:

a)      This paper should be placed next correctly with the paper titled “Generation of Cyberattacks Leading to Safety Top Event Using AltaRica: an Automotive Case Study” of the same co-authors to show the work progress and avoid incorrect self-plagiarism.

b)      Figure 1 draws the architecture of the Autonomous Ship System. Such critical systems are often doubled to enrich its safety. How it is represented on the graph and in the event model? Also, on Figure 1, it seems that some subsystems can be connected to rather subsystems: e.g. ECDIS to GPS; SCC to SAT. It is recommended to double check the system architecture graph.

c)      Section 2.5 lists Hypotheses regarding the conditions of top events. Namely, Hypothesis 3 states the attacker penetrating the system from AIS or SAT, and not from both ones in single attack. In reality, attacks can be complex, multi-stepped and multi-targeted. How can the proposed model represent such kinds of attacks? The Hypothesis is limited to 2 subsystems (AIS and SAT), and what about any penetrations through other subsystems?

d)      The abstract states that the authors “… present an experiment conducted with discrete event simulation to analyze the effects of multi-step cyberattacks.” But string #178 considers only “atomic” attacks involving only one component or only one link”. It is not clear what multi-step or atomic attacks are modelled?

e)      Section 4 does not provide experimental results except AltaRica code. The experimental outcome should present the merit of the proposed technique: e.g. new attacks were founded, new routes of the attacks were discovered, the formal verification goes quicker than the alternatives, etc. all that can help people to understand the novelty and merit of the proposed technique.

f)       Section 5.1 just reviews the related works. But the most expected output of this section is the comparison of the features and characteristics of the proposed technique against the concurrent solutions. There should be highlighted the differences and advantages of the authors’ approach.

g)      It is recommended to unite Sections 5.2 and 6, merging the Conclusion of the work and setting the future work plan.

The manuscript is recommended to be accepted, after light revision.

Reviewer 2 Report

The manuscript presents an experiment to analyze security risk assessment of cyber-physical system (CPS) using discrete event systems to analyze effects of cyber-attacks on CPS. The comments are as follows.

* The manuscript has presented experiment under constraint environments. However, in real world, the situation is quite complex and could possibly having unconstraint environments.

* Modeling of attacks are useful to study primacies of experiments/situations under consideration which is very much simple and predetermined.

* Novelty and contributions are limited. 

Reviewer 3 Report

1- the paper no have any sound idea 

2- the abstract has no control over real problems as cyber attacks on physical 

3-   no related work for comparison  and summary 

4- no method and algorithm and step for proposal idea 

5- where is the result of this serial about figures 

6- how can other authors use this idea 

7- must application in the real area 

8- need more development